# Effect of Acetylated SEBS/PP for Potential HVAC Cable Insulation

**DOI:** 10.3390/ma14081811

**Published:** 2021-04-07

**Authors:** Peng Zhang, Xuan Wang, Jiaming Yang, Yongqi Zhang

**Affiliations:** 1Key Laboratory of Engineering Dielectrics and Its Application, Ministry of Education, Harbin University of Science and Technology, Harbin 150080, China; zp199620@163.com (P.Z.); zhangyongqihust@163.com (Y.Z.); 2School of Electrical and Electronic Engineering, Harbin University of Science and Technology, Harbin 150080, China

**Keywords:** polypropylene, voltage stabilizer, electrical tree, thermoplastic elastomer, breakdown field strength

## Abstract

Blending polypropylene (PP) with thermoplastic elastomer SEBS can effectively improve the mechanical toughness of PP, thus leading to the promise of SEBS/PP as the primary insulation material for high voltage alternating current (HVAC) cables. However, the growth of electrical trees during cable operation limits the application of SEBS/PP. In this paper, acetylation reaction is used to construct acetophenone group at the end of the benzene ring on SEBS so that it has the effect of both a toughening agent and a voltage stabilizer. Then PP was melt blended with acetylated SEBS (Ac-SEBS), and the effects of Ac-SEBS on the mechanical properties, electrical tree resistance, alternating current (AC) breakdown strength, and dielectric spectrum of PP were mainly investigated with reference to PP and SEBS/PP. The results showed that Ac-SEBS with 30% content could enhance the mechanical toughness of PP and improve the electrical tree resistance and AC breakdown strength of SEBS/PP. The AC breakdown field strength of Ac-SEBS/PP reached the highest when the acetylation level was 4.6%, which was 9.2% higher than that of SEBS/PP. At this time, Ac-SEBS was also able to absorb high-energy electrons through the keto-enol interchange isomerization reaction, which inhibited the initiation and growth of electric trees and caused the development of electric dendrites in a jungle-like manner. Moreover, the dielectric loss factor of AC-SEBS/PP in power frequency is within the allowable range of industry. Therefore, Ac-SEBS/PP is expected to be applied to HVAC cables, thus further improving the efficiency of HVAC power transmission.

## 1. Introduction

Cross-linked polyethylene (XLPE) is widely used as the primary insulation material for power cables because of its excellent electrical and mechanical properties. However, the production of XLPE cables generates a large number of byproducts, such as cumyl alcohol and methane, which tend to aggravate insulation aging under high electric fields [1,2,3]. In addition, the cross-linking process will result in XLPE being a thermoset material that is extremely difficult to recycle and degrade [4]. The incineration of XLPE will produce large amounts of carbon dioxide, causing a serious greenhouse effect. Therefore, the research on new cable insulation materials that can replace XLPE has attracted much attention.

Polypropylene (PP) is a thermoplastic insulation material, which can be recycled when the cable life expires [5,6]. Therefore, there is no need to incinerate used PP, thus meeting environmental needs. Compared with XLPE, the production of PP cables can effectively reduce impurities generation because no cross-linking process is required. Besides, PP also has excellent insulation performance, chemical resistance, and high-temperature resistance. Therefore, many researchers consider PP promising to replace XLPE as the primary insulation for power cables [7,8]. However, PP suffers from a lack of flexibility and poor low-temperature toughness. The mechanical toughness of PP as high voltage alternating current (HVAC) material must be improved to meet the requirements of cable laying and installation [9,10]. Numerous studies have proven that thermoplastic elastomers are modified additives that can significantly enhance the toughness of PP [11,12,13]. In addition to the defects in mechanical properties, the growth of electrical trees in polymeric insulating materials under alternating current (AC) electric fields limits its applications. The electrical tree is a micron-level discharge destruction channel formed by local electric field concentration caused by impurity, bubble, and other defects in insulating materials, which leads to local breakdown. Once electrical trees are generated, they develop rapidly in the form of dendrites and eventually lead to the breakdown of insulating materials, which poses a significant threat to the safe operation of cables [14,15,16,17]. Therefore, enhancing the electrical resistance of PP/elastomer is an urgent problem to be solved.

Inorganic nanofillers and voltage stabilizers have been reported to be widely used to improve the breakdown strength and electrical tree initiation voltage of polymers and inhibit the growth of electrical trees [18,19,20,21,22]. Chi et al. [23] found that the addition of nano- montmorillonite to polyethylene (PE) and PP inhibited the growth of electrical trees. Gao et al. [24] added nano-SiO2 to PP/SEBS and found that the initial voltage and growth of the electrical tree of the composites were reduced. However, to ensure high dispersion of nanoparticles in the polymer matrix, proper surface modification of the nanoparticles, and long mixing time of the composites are required. The surface modification affects the dielectric properties of the composites [25]. PP is susceptible to degradation after a long period of blending. In addition, the long-term stability of nanocomposites has not been proven. Therefore, nanofillers cannot be directly applied in cable production at present.

The primary function of voltage stabilizers is to inhibit the electrical trees of insulating materials under the long-term electric fields and, thus, to improve the electrical resistance of insulating materials [26]. Most voltage stabilizers are aromatic compounds, such as acetophenone or benzyl. Aromatic compounds have high electron affinity energy and can collide with high-energy electrons under strong electric fields, absorbing the energy of high-energy electrons through excitation, thus reducing the damage of high-energy electrons to polymer molecular chains, which, in turn, improves the electrical resistance of insulating materials [27,28,29]. Jarvid et al. [30] showed that the addition of benzoyls to XLPE could increase its electrical tree initiation voltage by up to 70%. However, small molecules acetophenone are poorly compatible with polymers, and migration of acetophenone can affect the long-term operation of cables. Zhang et al. [31,32] proposed that acetophenone and its derivatives absorb high-energy electron energy through keto-enol exchange isomerization reaction, thus reducing the damage of high-energy electrons to XLPE molecular chains. Additionally, acetophenone bonding to macromolecular chains can also improve the electrical tree resistance of insulating materials. Therefore, in order to improve the poor compatibility of polymers with acetophenone, the grafting of voltage stabilizers has attracted attention. Dong et al. [33] prepared XLPE with acetophenone structural units, which inhibited the growth of electrical trees and the migration of acetophenone. However, at present, there are few reports on whether voltage stabilizers containing acetophenone structural units can improve the AC insulation properties of PP/elastomer composites, especially the inhibitory effect on the initiation and growth of electrical trees.

SEBS is a new thermoplastic elastomer with excellent heat resistance and is commonly used for toughening modification of PP because of its good compatibility with PP. Based on this, this paper investigates the effect of acetophenone voltage stabilizers on the AC insulation performance of PP/SEBS. In order to inhibit the migration of acetophenone, the acetylation reaction was used to construct acetophenone structural units on the benzene ring at the end of SEBS, and PP was melt-blended with acetylated SEBS (Ac-SEBS). The electrical tree resistance, mechanical properties, AC breakdown strength, and dielectric spectrum of Ac-SEBS/PP were tested separately. The study shows that the introduction of Ac-SEBS can enhance the breakdown strength and electrical tree initiation voltage of the SEBS/PP, and inhibit the growth of electric trees and maintain the toughening function of the elastomer. The effect of Ac-SEBS on PP is still within the permissible range for engineering applications at the power frequency voltage, which does not affect the use of the material at the power frequency voltage. Ac-SEBS/PP is expected to be applied to HVAC cables, thus further improving HVAC power transmission efficiency.

## 2. Materials and Methods

### 2.1. Materials

The materials selected for the preparation of composites include homopolymer PP (T30S, Sinopec, Beijing, China), thermoplastic elastomer SEBS (g1652, Kraton), methylene chloride (Tianjin Fuyu Fine Chemical Co., Tianjin, China), acetyl chloride (Aladdin industrial corporation, Shanghai, China), anhydrous aluminum chloride (Aladdin industrial corporation, Shanghai, China), methanol (Tianjin Fuyu Fine Chemical Co., Ltd, Tianjin, China), n-heptane (Tianjin Fuyu Fine Chemical Co., Ltd, Tianjin, China), and Antioxidant 1010 (Dongguan Shanyi Plasticizing, Dongguan, China).

According to the Friedel–Craft reaction, the synthesis route of Ac-SEBS is shown in Figure 1. A total of 10 g of SEBS was poured into 200 mL of methylene chloride. After it was all dissolved, 5 mL of acetyl chloride was poured into the solution and stirred for 5 min, and, after it was stirred well, a certain mass (0.4, 0.5, 0.6, 0.7 g) of anhydrous aluminum chloride was added to the solution. The reaction was carried out at room temperature for 3 h. After the reaction, 300 mL methanol was poured into the reaction solution to precipitate white flocculent precipitates. The precipitates were filtered and put in a vacuum oven at 50 °C for drying for 5 h. The dried precipitates were poured into 200 mL of methylene chloride again, and after they were all dissolved, 300 mL of methanol was poured into the solution to precipitate white flocculent precipitates, and the precipitates were filtered dried in a vacuum oven at 50 °C for 5 h to obtain relatively Ac-SEBS. In this experiment, the acetylation degree of SEBS was changed by changing the addition of anhydrous aluminum chloride.

PP, 30% Ac-SEBS, and 1% antioxidant were added to the torque rheometer, and the melting process was carried out at 190 °C with a rotor speed of 60 rpm for 5 min. After taking out the homogeneously mixed specimens, the flat vulcanizer was applied to press the samples at a temperature of 190 °C and a pressure of 15 MPa to obtain samples of different thicknesses according to the needs of the experiment.

### 2.2. Characterization and Testing Scheme

The acetylation results were examined by measuring the Fourier transform infrared (FT-IR) spectra of SEBS and Ac-SEBS on a Nicolet iS5 spectrometer (Madison, WI, USA). The measurements were performed in the range of 400 to 4000 cm^−1^ with a resolution of 4 cm^−1^. The degree of acetylation of Ac-SEBS was determined by the nuclear magnetic resonance hydrogen (^1^H NMR) on a Bruker AVANCE III (Karlsruhe, Germany) at 400 MHz (^1^H), with CDCl_3_ as the solvent and Tetramethylsilane (TMS) as the internal standard. For observing the microscopic morphology of the Ac-SEBS/PP composite system, specimens fractured by liquid nitrogen were immersed in n-heptane for 30 min. After washing and drying the specimens with alcohol, the etched cross-sections were sprayed with gold and then imaged by a scanning electron microscopy (SEM, SU8020, Tokyo, Japan).

Tensile tests were performed using a tensile testing machine (CMT6000) with a fixed speed of 50 mm/min at 25 °C. A total of 5 specimens of each material were prepared for repeated measurements. The specimens were dumbbell-shaped with a thickness of 1 mm, a width of 4 mm, and a gauge length of 20 mm. The breakdown test was performed using an AC frequency breakdown system with a uniform ramping rate of 1 kV/s according to ASTM D 149-97a (2004) [34]. The samples with a thickness of 50 mm were placed between two cylindrical electrodes and all immersed in silicone oil to prevent surface flashover. The high voltage electrode diameter was 25 mm, and the diameter of the ground electrode was 75 mm. After the breakdown of the sample, the breakdown voltage U and the thickness of the breakdown point d were recorded. According to the formula E = U/d, the breakdown field strength of the sample was calculated. 12 samples of each material were tested, and the results were processed using the two-parameter Weibull statistical distribution.

The needle-plate electrode system was used for the initiation and growth of electrical trees, and the distance between the needle and plate was about 3 mm. During the experiment, the samples were immersed in silicone oil, 10 samples of each material were selected as a group, and the voltage was increased at a constant speed at a speed of 100 V/s. The morphology of the electrical tree was observed by an optical system consisting of a charge-coupled device (CCD) camera and an optical microscope (Suzhou, China), and the system was accessed through the computer interface. When the growth length of the electrical tree branch was observed to reach 10 μm, the voltage at this point was recorded as the electrical tree initiation voltage. During the test of the growth of electrical trees, 7 kV AC power frequency voltage was applied to the sample for 120 min. The dielectric properties of the specimens with a thickness of 200 μm were tested on a broadband dielectric spectrometer (Alpha-A, Frankfurt, Germany) in the frequency range of 1 to 10^6^ Hz. Before the test, round aluminum electrodes of 25 mm were plated on both sides of the sample.

## 3. Results and Discussion

### 3.1. Structural Characterization

The infrared spectra of SEBS and Ac-SEBS are given in Figure 2 to demonstrate the chemical structures of SEBS and Ac-SEBS. Compared with the spectra of SEBS, new absorption peaks appear at 1684, 1269, and 826 cm^−1^ in Ac-SEBS. The absorption peak at 1684 cm^−1^ identifies the vibration of C=O stretching. The absorption peak located at 1269 cm^−1^ may derive from the vibration of the aromatic ketone skeleton. The absorption peak located at 826 cm^−1^ identifies the para substitution of the benzene ring for the C-H surface bending vibration [35]. Therefore, the IR spectrograms can prove that the acetophenone structural unit is constructed at the end of the benzene ring of Ac-SEBS.

The degree of acetylation of Ac-SEBS was calculated by testing the ^1^H NMR spectra to examine the effect of different aluminum chloride additions on the degree of acetylation. Figure 3 shows the ^1^H NMR spectra of different Ac-SEBS. In the ^1^H NMR spectra, the chemical shifts of the resonance of benzene ring protons are generally from 6 to 8 ppm. Therefore, it can be found from the ^1^H NMR spectra of Ac-SEBS that the resonance of benzene ring proton occurred in three regions (A, 6.3–6.8 ppm, B, 6.8–7.2 ppm, and C, 7.35–7.8 ppm). Region A represents the ortho proton peak of the benzene ring on SEBS, and region B represents the meta and para proton absorption peaks of the benzene ring in SEBS. After acetylation of SEBS, the carbonyl group is generated on its benzene ring. The carbonyl group undergoes π-π conjugation with the double bond on the benzene ring. The two absorption peaks adjacent to the carbonyl group on the benzene ring migrate to the lower field so that region C can be observed on the ^1^H NMR spectra of Ac-SEBS. Region D (2.4–2.6 ppm) represents the proton peak of the methyl group on the acetophenone structural unit produced by the acetylation of SEBS [35]. The area of the resonance peak in the ^1^H NMR spectrum is proportional to the number of protons producing this peak, so the degree of acetylation of Ac-SEBS can be calculated based on the area ratio of region A to region D. The degree of acetylation of the sample could be calculated by Equation (1):(1)$$\mathrm{sub}\%=\frac{2\times A_{D}}{3\times A_{A}}\times 100\%$$
where Sub is the degree of acetylation of Ac-SEBS, A_A_ is the proton peak area of A, and A_D_ is the proton peak area of D.

Figure 4 shows the microstructures of PP and its composites imaged by SEM. Compared with the morphology of pure PP, the pores in the morphology of the composite samples are formed by the n-heptane etching of Ac-SEBS dispersed in PP. The SEBS and Ac-SEBS in the composite are distributed in the sea-island structure. In the PP matrix, the number of SEBS pores is minimal, the size is small, and the distribution is very uniform, and there is no apparent agglomeration phenomenon, which indicates that PP and SEBS are not entirely compatible, but they are extremely compatible. After the introduction of Ac-SEBS in the PP matrix, the number and size of pores increased, but when the degree of acetylation was low, Ac-SEBS was still uniformly distributed in the PP matrix, and with the increase in the degree of acetylation, the agglomeration phenomenon of Ac-SEBS appeared in the PP matrix. This is because when the acetylation degree of Ac-SEBS is higher, Ac-SEBS has higher molecular weight and more polar groups, and the difference between the molecular weight of PP and Ac-SEBS and the interaction of polar groups simultaneously lead to the poor compatibility of PP and Ac-SEBS, which finally leads to the poor dispersion of Ac-SEBS in the PP matrix.

### 3.2. Stress-Strain Curve

Mechanical toughness is an important index to judge whether a cable can be used. Figure 5 shows the tensile stress-strain curves of PP and its composites. Their corresponding tensile strength and elongation at break are shown in Table 1. It can be seen from the figure that the tensile strength and elongation at break of SEBS/PP and Ac-SEBS/PP are much higher than those of PP. This is because SEBS is a highly flexible material, and SEBS and Ac-SEBS are dispersed in PP in the form of the dispersed phase, and when the composites are subjected to external force, the area where the elastic particles are located will generate a large shear strain, which will cause energy consumption and thus playing a toughening effect and increasing the elongation at break of the composites. However, with the increase in the degree of acetylation, the dispersion of the elastomer becomes worse, and the effect of energy consumption is weakened, leading to a gradual decrease in elongation at break. It can also be seen from the figure that the tensile strength of Ac-SEBS/PP is smaller compared with that of SEBS/PP, but it gradually increases with the increase in the degree of acetylation. It is possible that the higher tensile strength of SEBS/PP is due to the stronger entanglement between PP and SEBS molecular chains. However, the introduction of Branched chains on the polystyrene block of Ac-SEBS increases the intermolecular distance, resulting in a decrease in the intermolecular force and a decrease in tensile strength. With the introduction of more acetyl groups, the molecular weight of Ac-SEBS increased, resulting in the increase in intermolecular force of Ac-SEBS/PP, and the tensile strength of Ac-SEBS/PP increased with the increase in acetylation degree. According to reference [36], the tensile strength and elongation at break of XLPE are 23.6 MPa and 579%, respectively. Based on this, it can be found that the tensile strength and elongation at break of Ac-SEBS/PP are higher than those of XLPE when the degree of acetylation is low. Therefore, Ac-SEBS/PP has the potential to replace the current XLPE.

### 3.3. AC Breakdown Strength

Figure 6 shows the Weibull distribution of the AC breakdown field strength of PP and its composites, where the shape parameter represents the degree of data diffusion and the scale parameter indicates the AC breakdown field strength in the Weibull distribution where the cumulative breakdown probability of specimens reaches 63.2%. As shown in Figure 6, the AC breakdown field strength of PP is 146.6 kV/mm, and the introduction of a large amount of SEBS reduces the AC breakdown strength of PP/SEBS to 137.3 kV/mm with a decrease of 6.3%. This is because SEBS/PP with the sea-island structure may have micropores, which will aggravate the partial discharge of PP under AC voltage and increase the free path of electrons, making it easy to accumulate high-energy electrons. On the other hand, the Polystyrene (PS) segment of SEBS is microscopically phase-separated from the PE segment, and the addition of SEBS to PP will introduce a large number of interfaces, causing more serious interfacial losses and causing a decrease in the AC breakdown strength of PP. After acetylation of SEBS, the AC breakdown field strength of SEBS/PP could be improved. However, with the increase in acetylation degree, the AC breakdown field strength of the composite first increased and then decreased. The highest breakdown field strength of the composite was achieved when the acetylation degree was 4.6%, which increased by 9.2% compared with PP/SEBS. The increase in AC breakdown field strength is mainly attributed to the acetophenone structural unit on Ac-SEBS. The keto-enol interchange isomerization reaction of the acetophenone group can consume the energy of electrons and inhibit the damage of the molecular chain by high-energy electrons, thus improving the breakdown strength of the composite. In addition, the scattering effect of dipoles on Ac-SEBS on electrons also positively affects the improvement of AC breakdown field strength. However, when the acetylation degree is too high, the Ac-SEBS agglomerates in the PP matrix, and although the acetophenone structural unit can absorb electrons, the more serious interfacial loss and partial discharge effects play a dominant role on the breakdown field strength, which finally causes the AC breakdown field strength of the composite system to decrease. These results indicate that Ac-SEBS with a suitable degree of acetylation can improve the AC breakdown field strength of SEBS/PP. According to reference [37], the AC breakdown field strength of XLPE is 87.74 kV/mm. Accordingly, it can be found that the electrical strength of Ac-SEBS/PP is better than that of XLPE.

### 3.4. Electrical Tree Characteristics

Figure 7 shows the Weibull distribution of the electrical tree initiation voltage of PP and its composites. The shape parameter represents the degree of data diffusion, and the scale parameter represents the electrical tree initiation voltage. From the Figure, it can be found that the introduction of SEBS decreased the electrical tree initiation voltage of PP, and the electrical tree initiation voltage of Ac-SEBS/PP was higher than that of SEBS/PP, and the electrical tree initiation voltage of Ac-SEBS/PP increased first and then decreased with the increase in the acetylation degree. When the electrical tree initiation voltage of Ac-SEBS/PP reaches the maximum, it is 37.1% higher than that of SEBS/PP. The mechanism of the influence of SEBS and Ac-SEBS on the electrical tree initiation voltage of PP is similar to that of the AC breakdown field strength. The sea-island structure of SEBS/PP may cause micropores in the composite system, thereby aggravating the partial discharge phenomenon in PP and promoting the initiation of electrical trees. Ac-SEBS completes the keto-enol isomerization reaction by absorbing high-energy electrons, which reduces the possibility of damage to the PP molecular chain, thereby inhibiting the initiation of electrical branches. When the degree of acetylation is too high, Ac-SEBS clusters in the PP matrix, causing severe partial discharge, which makes the electrical tree easier to be initiated. According to reference [37], the electrical tree initiation voltage of XLPE is 5.99 kV. Accordingly, it can be found that the electrical tree resistance of Ac-SEBS/PP is better than that of XLPE.

In order to judge the influence of the voltage stabilizer on the growth rate of the electrical tree, the length of the electrical tree was recorded at different times of pressure. The maximum vertical diffusion distance of electrical trees along the tip was defined as the length of electrical trees, and the average electrical dendrite length of each group of 10 specimens was used as the reference electrical dendrite length in this paper. The relationship between the tree length and the treeing time is illustrated in Figure 8. It is seen from Figure 8 that the electrical trees in SEBS/PP grew faster than PP, and the length of electric dendrites in SEBS/PP was longer when the voltage is applied for 120 min. Moreover, Ac-SEBS was obviously able to slow down the growth rate of electric dendrites. When the acetylation of Ac-SEBS reached 4.6%, the length of electrical trees in Ac-SEBS/PP almost did not change with time. However, with the further increase in acetylation of Ac-SEBS, the growth rate of electrical trees in the composite system accelerated, and when the degree of acetylation of Ac-SEBS reached 12%, the growth rate of electrical trees in the composites was almost the same as the growth rate of electric dendrites in SEBS/PP, and its length of electrical trees was much longer than that of SEBS/PP. Therefore, it can be concluded that a large amount of SEBS would promote the growth of electrical trees in PP, and Ac-SEBS can significantly inhibit the growth of electrical trees in the PP matrix when Ac-SEBS does not undergo serious agglomeration in the PP matrix. Its inhibition mechanism is consistent with the inhibition of the initiation of electrical trees.

Figure 9 shows the electrical tree morphology characteristics of the sample when the voltage is applied for about 7200 s. It can be seen from the figure that in the PP, the electrical trees develop along the two main carbonization channels, and the overall structure is relatively simple, while some branches become dense during the development. In the SEBS/PP composite system, the electrical trees developed along with multiple directions, and the structure of the electrical trees becomes complicated. The branches of SEBS/PP were dispersed, and the electrical trees with multiple carbonization channels are formed. According to the different degrees of acetylation, different shapes of electrical trees appeared inside Ac-SEBS/PP. When the degree of acetylation was 2%, the electrical trees of Ac-SEBS/PP had a relatively uniform and dense structure. Compared with SEBS/PP, its electrical trees developed fewer main channels, and the branches clustered together. When the acetylation degree was 4.6%, the electrical trees grown within Ac-SEBS/PP showed a denser jungle-like electrical tree, and the electrical trees developed in clusters. With the further increase in the degree of acetylation, the electrical trees of Ac-SEBS/PP developed in multiple directions again, and the branches became dispersed. The width of the electrical trees grown within Ac-SEBS/PP increased significantly, and the morphology of the electrical branches was very complicated.

It is analyzed that PP is a non-polar polymer with a regular structure, and its internal electrical weak areas are less, which cannot provide more channels for the growth of electrical trees. SEBS is not fully compatible with PP, and the introduction of SEBS in the PP matrix will destroy the regular structure of PP and introduce a large number of interfaces, which provides more electrical weak areas for the development of electrical trees and increases the randomness of electrical trees growth. Moreover, the insulating property of SEBS is not as good as that of PP, which will make it easier for the electrical trees to form branches and develop rapidly. While Ac-SEBS has a high electron affinity, it will be impacted by electrons first under the strong electric field, and the acetophenone group consumes the energy of electrons through keto-enol interchange isomerization reaction and releases it with relatively harmless energy such as luminescence or vibration. When acetylation is not high, Ac-SEBS is uniformly distributed in the PP matrix in the form of the voltage stabilizer, which achieves the complementation of electrically weak areas and thus reduces the main development channel of electrical trees. In addition, when the electrical trees developed to the vicinity of Ac-SEBS, Ac-SEBS caused the electrical trees to change their development direction by absorbing electrons and develop along with the electrically weak areas. The well-dispersed Ac-SEBS forces the branches of the electrical trees to cluster together to form a dense jungle-like electrical tree eventually. When the degree of acetylation is high, the Ac-SEBS appears to be agglomerated, increasing the interfacial area in the composite system and forming more physical or chemical defects. This leads to an increase in the average free travel of electrons inside the composite system, which intensifies the damage to the PP molecular chains, thus facilitating the expansion and extension of the electrical tree channels and providing more channels for the growth of electrical trees.

### 3.5. Dielectric Properties

Figure 10 shows the dielectric constant curve of PP and its composites. Figure 11 shows the dielectric loss curve of PP and its composites. The dielectric constant spectra show that the dielectric constant of all samples does not change significantly in the test frequency range, and the addition of SEBS improves the dielectric constants of the composites. With the increase in acetylation degree, the dielectric constant first decreased and then increased. PP is a non-polar dielectric, and its primary polarization mechanism is electron displacement polarization. The establishment time of electron displacement polarization is very short, about 10^−15^ to 10^−16^ s. Therefore, the relative dielectric constant of PP and its composites hardly change with frequency. However, SEBS is not fully compatible with PP, and the addition of SEBS to PP introduces a large number of interfaces, causing interfacial polarization and increasing its relative dielectric constant. With the introduction of the acetophenone group, the rotation of the benzene ring branched chains becomes difficult. At the same time, Ac-SEBS exists in the PP matrix, which restricts the movement of the PP molecular chain, and the turning-direction polarization is difficult to happen. The increase in acetyl groups has a more apparent inhibitory effect on the dipole turning-direction polarization. Therefore, the relative dielectric constant of Ac-SEBS/PP decreases with the increase in acetylation degree. However, when the acetylation degree reaches 12%, Ac-SEBS is exceptionally incompatible with PP, which intensifies the interfacial polarization and causes an increase in the relative dielectric constant.

In the dielectric loss spectrum, when the frequency is low, all polarizations can be established, and the loss factor decreases as the frequency increases. As the frequency approaches the relaxation region, the dielectric loss factor increases. Ac-SEBS/PP composites enter the relaxation region in the lower frequency, which indicates that there is a relaxation polarization mechanism that has a significant influence on the loss factor of Ac-SEBS/PP. This relaxation polarization mechanism may come from the interface polarization caused by poor material compatibility. With the increase in acetylation, the dielectric loss factor increases, which is due to the inhibitory effect of Ac-SEBS on the movement of PP chains, which tends to cause intermolecular collisions and friction in the low-frequency region. The increase in benzene rings with acetyl groups has a more substantial inhibitory effect on the molecular chains, and the increase in intermolecular collisions and friction is more likely to cause losses. In addition, the introduction of a large number of interfaces will increase the interfacial loss of the material. Therefore, the dielectric loss factor increases with the increase in acetylation. At a voltage frequency of 60 Hz, the loss factor of both PP and its composites did not exceed 0.001, so the effect of Ac-SEBS on PP is within the acceptable engineering range.

## 4. Conclusions

In this study, the possibility of applying Ac-SEBS/PP to HVAC cables was investigated. Furthermore, the mechanical properties, electrical strength resistance, and dielectric spectrum were mainly investigated. It can be found that Ac-SEBS has good compatibility with PP when the degree of acetylation is not high. The blending of Ac-SEBS with 30% content into PP can effectively improve the tensile strength and elongation at break of PP to meet the mechanical property requirements of HVAC cables. Ac-SEBS with high electron affinity can absorb high-energy electrons and consume the energy of electrons through the keto-enol interchange isomerization reaction of the acetophenone group to improve the electrical resistance of PP. When the degree of acetylation reached 4.6%, the AC breakdown field strength of Ac-SEBS/PP was increased by 9.2% compared with that of SEBS/PP. Moreover, Ac-SEBS was able to make the electrical trees within PP develop in a jungle-like manner and inhibit the initiation and growth of electrical trees. Besides, the dielectric loss factor of Ac-SEBS/PP does not exceed 0.001 at the power frequency voltage, which is still within the permissible range for engineering applications and does not affect the use of the material at the power frequency voltage. According to the experimental results, it can be concluded that Ac-SEBS/PP can meet the mechanical toughness and electrical strength requirements of HV cables when the degree of acetylation is appropriate. In addition, the mechanical properties and electrical strength of Ac-SEBS/PP are better than XLPE. Therefore, the Ac-SEBS/PP composite system is a promising candidate for HVAC cable insulation.

## Figures and Tables

**Figure 1 materials-14-01811-f001:**
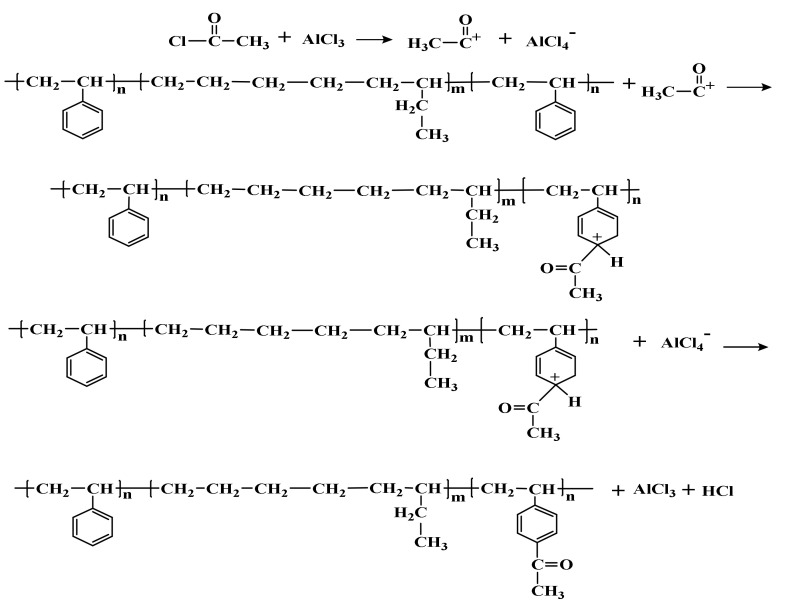
The preparation of Ac-SEBS.

**Figure 2 materials-14-01811-f002:**
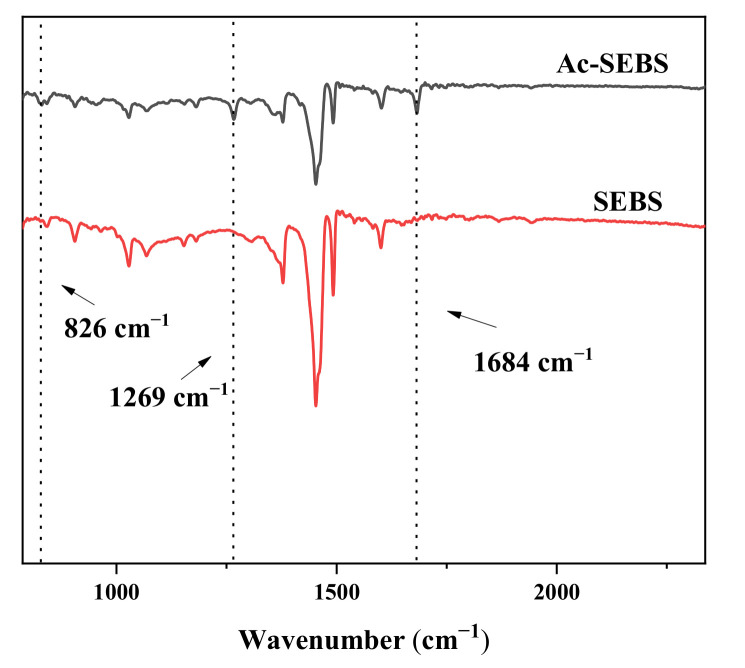
Infrared spectra of SEBS and Ac-SEBS.

**Figure 3 materials-14-01811-f003:**
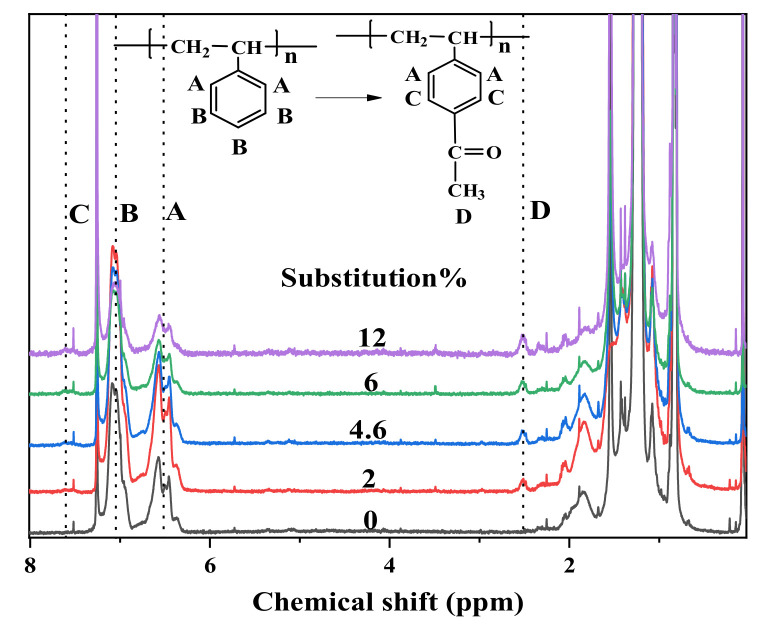
^1^H NMR spectra of SEBS and Ac-SEBS.

**Figure 4 materials-14-01811-f004:**
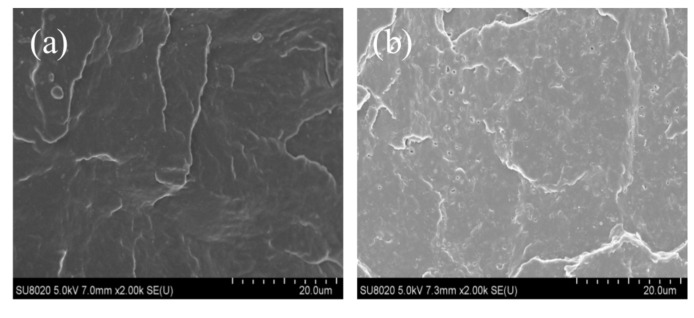
SEM images of Ac-SEBS/PP composites with different acetylation degrees of SEBS: (**a**): PP; (**b**): 0%; (**c**): 2%; (**d**): 4.6%; (**e**): 6%; (**f**): 12%.

**Figure 5 materials-14-01811-f005:**
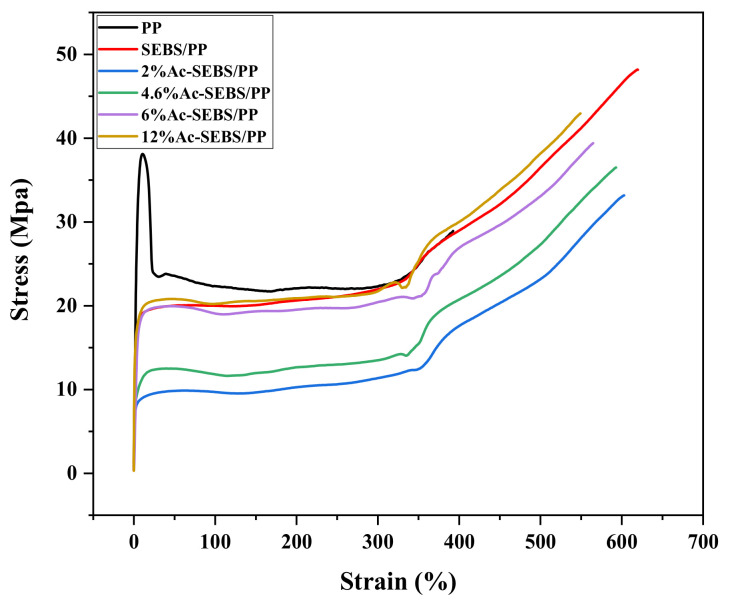
Stress-strain curve of PP and its composites.

**Figure 6 materials-14-01811-f006:**
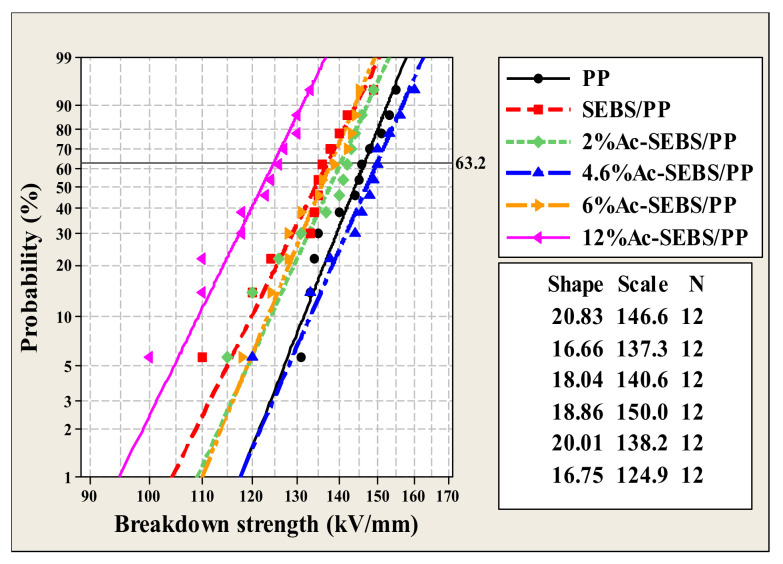
AC breakdown field strength of PP and its composites.

**Figure 7 materials-14-01811-f007:**
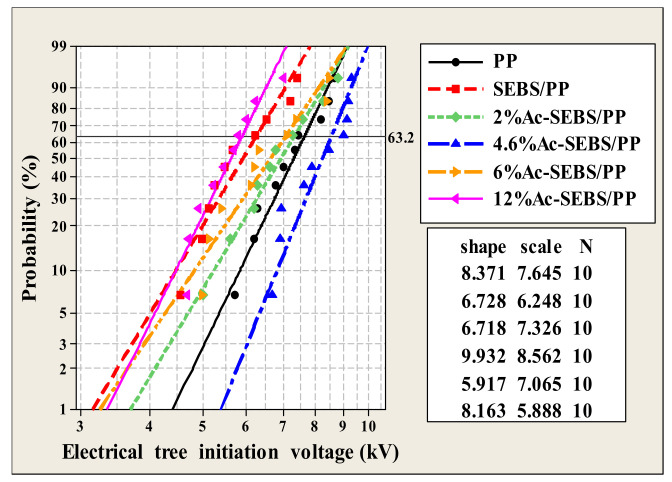
Weibull distribution of the electrical tree initiation voltage of PP and its composites.

**Figure 8 materials-14-01811-f008:**
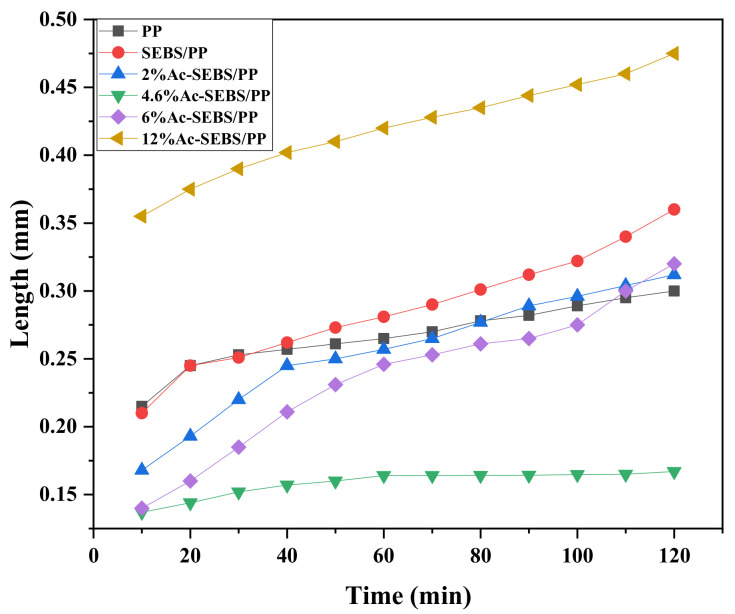
Relationship between the tree length and the treeing time.

**Figure 9 materials-14-01811-f009:**
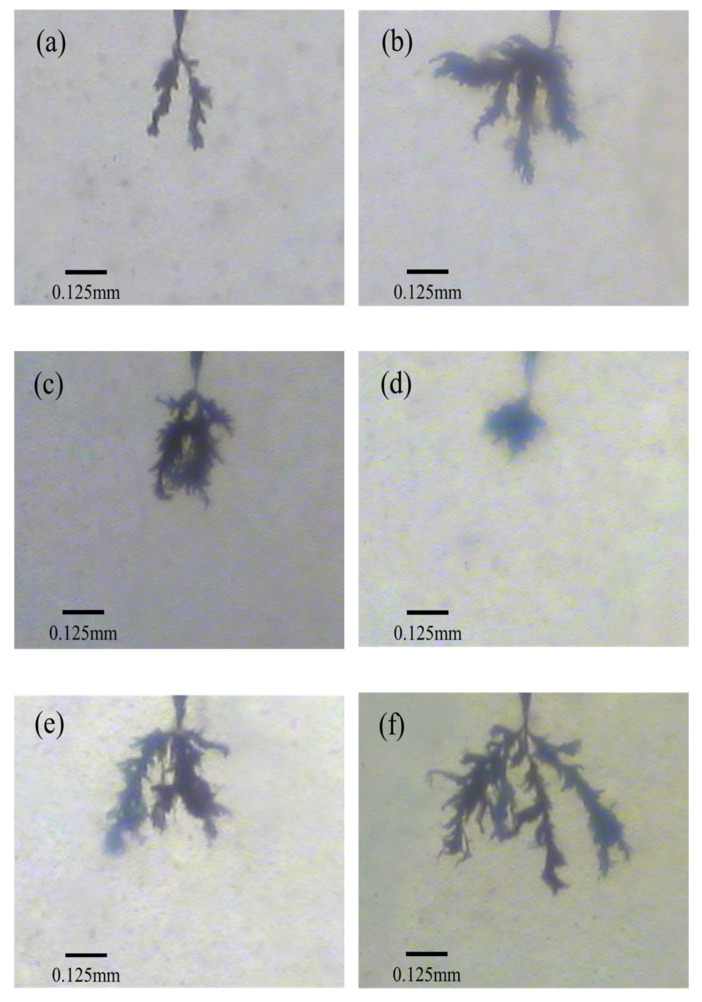
The electrical tree microscopic morphology of PP and Ac-SEBS/PP with different degrees of acetylation when the voltage is applied for 7200 s (**a**): PP, (**b**): PP/SEBS; (**c**): 2%; (**d**): 4.6%; (**e**): 6%; (**f**): 12%.

**Figure 10 materials-14-01811-f010:**
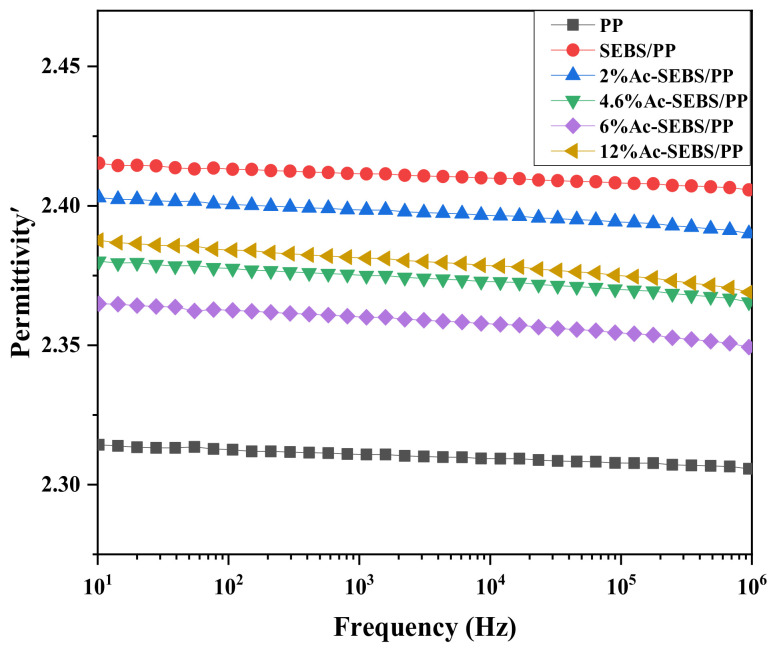
The dielectric constant spectrum of PP and its composites.

**Figure 11 materials-14-01811-f011:**
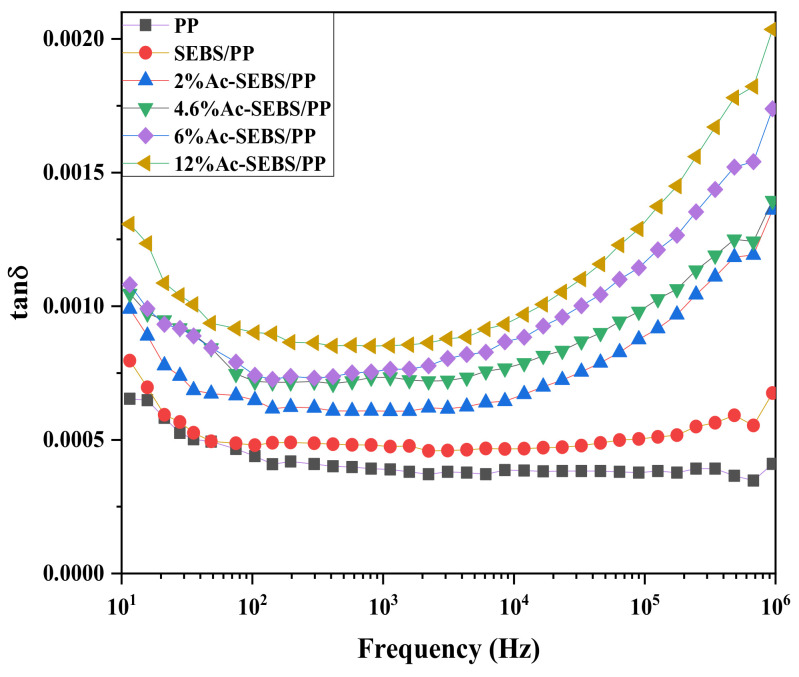
Dielectric loss spectrum of PP and its composite materials.

**Table 1 materials-14-01811-t001:** Summary of tensile test data of the samples.

Sample	Elongation at Break (%)	Tensile Strength (MPa)
PP SEBS/PP 2% Ac-SEBS/PP 4.6% Ac-SEBS/PP 6% Ac-SEBS/PP 12% Ac-SEBS/PP	392.60 ± 38.38 619.75 ± 39.15 602.86 ± 29.56 593.01 ± 32.32 564.12 ± 34.85 549.25 ± 34.40	28.92 ± 2.15 47.15 ± 1.76 33.17 ± 1.84 36.50 ± 1.73 39.39 ± 1.69 42.95 ± 2.02

## Data Availability

The data presented in this study are available on request from the corresponding author.

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
