# Peer review of "Effect of Acetylated SEBS/PP for Potential HVAC Cable Insulation"

_materials, 2021, doi:10.3390/ma14081811_

Round 1

Reviewer 1 Report

In this paper the authors investigate the effect acetylated SEBS on a wide variety of physical properties of Polypropylene. It was shown that Ac-SEBS/PP evaluates better mechanical and electrical characteristics in comparison to PP and SEBS/PP. The authors provide an extensive introduction with the necessary scientific background and a large number of references. The description of the experimental procedure is detailed as well as the presentation of the results.

My comments to the authors are the following

  1. Did you follow any standard test method for dielectric breakdown voltage of solid insulating materials such as ASTM D149-20?
  2. In line 154 of the text you mentioned that each material was tested 12 times. Did you test the same sample of each material 12 times or 12 different samples of each material? I think that this is not quite clear.
  3. In lines 159-160 you mentioned that the voltage application was stopped when the length of electric tree branch was observed to reach 10μm. How did you observe the growth of the tree inside the sample?
  4. I think it will be helpful the authors to add, if possible, some figures of the experimental setups for breakdown tests, dielectric properties tests and especially the experimental setup for recording the initiation and growth of electrical trees.

Reviewer 2 Report

The paper presents a study about an effect of acetylated SEBS/PP for high voltage AC cable. Authors say, that blending polypropylene, called PP, with thermoplastic elastomer SEBS is able improve mechanical properties of PP, what leads to promise of SEBS/PP as insulation material for high voltage cables. Anyway, the increasing electrical trees during cable operation may decrease application of SEBS/PP. Authors used acetylation reaction in order to construct acetophenone group at the end of the benzene ring on SEBS. Next, PP was melt blended with acetylated SEBS, and the effects of SEBS on the mechanical properties, electrical tree resistance, breakdown voltage, and dielectric spectrum of PP were studied to reference with PP and SEBS/PP. Obtained results proved that Ac-SEBS with 30% content is able to increase mechanical toughness of PP and increase electrical tree resistance.

Comments and questions:

  1. First chapter introduces to studied problem, what was effect of acetylated SEBS/PP for high voltage AC cable. Authors explain common materials used as high voltage AC insulation as cable insulation system, such as cross-linked polyethylene (XLPE), and PP. They indicate advantages and disadvantages of mentioned materials.
  2. Second chapter explain used in study materials and methods. Authors describe used composite various materials. They present materials properties, important from insulation point of view. Next, authors show characterization and testing scheme.
  3. Chapter 3 shows obtained results and made discussion. Authors explain structural characterization od prepared material. They present IR spectra of pure SEBS and prepared Ac-SEBS. They show 1H NMR spectra of SEBS and Ac-SEBS, and explain SEM images of Ac-SEBS/PP composites with different acetylation degrees of SEBS. Next, they present stress-strain curve, and to compare stress-strain curve of PP and its composites. Authors show obtained summary of tensile test data of the samples. Next, AC breakdown voltage was studied, and electrical tree characteristics were presented as Weibull distribution of the electrical tree initiation voltage of PP and its composites. Finally, authors showed relationship between the tree length and the treeing time. Next, they present results of investigations of dielectric properties, such as permittivity and tan(delta).

Reviewer 3 Report

  1. Page 1, paragraph 1, line 38; there are different kinds of pollution. Please provide more details on what kind of pollution and how it affects the ecosystem?
  2. Page 2, paragraph 2, line 40-42; what are the environment protection demands? Please state it.
  3. Paragraph 3, line 64-67; please clarify in the manuscript whether the production limitation is only due to dispersion of nanoparticles? Are there no other methods currently available that alleviates such shortcomings to uniformly disperse the nanoparticles?
  4. For figure 8, please mention how many times the experiment was repeated. Provide the y-axis, standard deviations (SD) in the graph and if the SD is too small, tabulate the data.
  5. The authors must provide a table comparing their research work with other published work within the last five years. The authors need to compare their work with other XLPE and PP with inorganic nanofillers and voltage stabilizers. Compare properties such electrical breakdown, stress/strain, electrical tree initiation voltage, etc.
  6. The authors mention in the introduction section that the PP suffers from a lack of flexibility and has poor low-temperature toughness. The authors have used Ac-SEBS in their work to overcome the shortcoming of other insulating materials. Although the authors have performed the stress/strain test, there is also no mention of the ambient testing conditions as well as no results on how the material property that the authors chose, would change its electrical breakdown, stress/strain, electrical tree initiation voltage, etc, when the ambient conditions (Temperature and Humidity) changes. If such insulating materials are going to be used for HVAC systems the authors must provide additional results to prove that the material works in varying ambient conditions (Temperature and Humidity). Manufacturers datasheets for insulated electrical wires always contain temperature rating, therefore such characterizations are essential. Without such characterizations concluding that “Ac-SEBS/PP composite system has great potential for application in HVAC cable” is inconclusive.

Round 2

Reviewer 3 Report

Comments have been addressed sufficiently.